# ANIME-READY: CONTROLLABLE 3D ANIME CHARACTER GENERATION WITH BODY-ALIGNED COMPONENT-WISE GARMENT MODELING

**Jiachen Qian[1,2], Hongye Yang[2], Youtian Lin[3], Tianhao Zhao[2], Feihu Zhang[2][\*], Yao Yao[3][†], Hengshuang Zhao[1]**
[1]The University of Hong Kong, [2]DreamTech, [3]Nanjing University

## ABSTRACT

Automated generation of 3D anime characters has become increasingly important in digital entertainment, including animation production, virtual reality, gaming, and virtual influencers. Compared to realistic human modeling, anime-style modeling requires exaggerated proportions, stylized surface details, and artistically consistent garments, posing unique challenges for automated 3D generation. Existing anime-style approaches often suffer from low-quality meshes, blurry textures, and lack of inner skeletons, which limits their usability in animation. In this work, we present a novel framework for high-quality 3D anime character generation to overcome these limitations by combining the expressive power of the Skinned Multi-Person Linear (SMPL) model with precise garment modeling. We extend the SMPL model to Anime-SMPL to better capture the distinctive features of anime characters, which enables unified skeleton generation and blendshape-based facial expression control, rendering the generated characters animation-ready. To complement the body model, we introduce a body-aligned component-wise garment generation pipeline, which models hairstyles, upper garments, lower garments, and accessories as structured components aligned with the underlying body geometry. Furthermore, our method produces high-quality skin and facial textures, as well as detailed garment textures, enhancing the visual fidelity of the generated characters. Experimental results demonstrate that our framework significantly outperforms baseline methods in terms of mesh quality, texture clarity, and garment-body alignment, making it well-suited for a wide range of applications in anime content creation and interactive media.

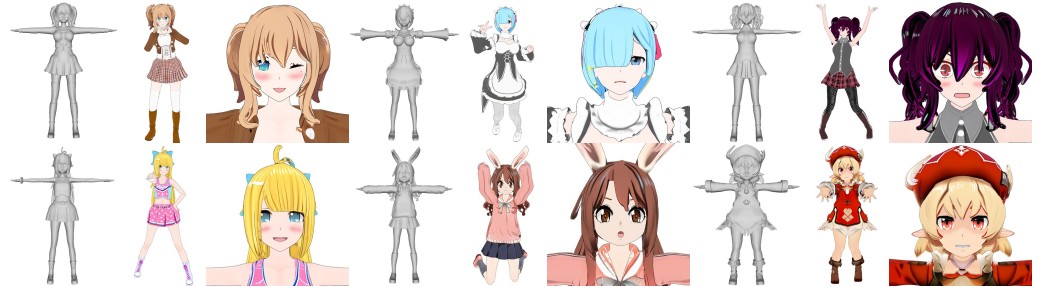

Figure 1: Our Anime-Ready generates high-quality, controllable 3D anime characters from text or a single image, with fine-grained control over actions such as finger movements and facial expressions.

[\*]Project leader.
[†]Corresponding author.
This research was supported by DreamTech, and the IP belongs to DreamTech.

# 1 INTRODUCTION

While recent advances in 3D human modeling have led to breakthroughs in reconstructing realistic human avatars, generating high-quality, animatable 3D characters in anime style remains a highly challenging task. Anime characters often feature stylized anatomy, typically exhibit more complex garments, diverse hairstyles, and exaggerated facial features, which challenges traditional reconstruction pipelines.

Recent methods (Peng et al., 2024; He et al., 2024b; Qian et al., 2025; Dong et al., 2025) reconstruct anime characters from images using large models (Hong et al., 2023), generating plausible geometry and textures. However, these approaches often fail to capture fine details, such as hand poses and hair structure. Besides, the results of these models are typically unrigged and lack consistent topologies, which is unsuitable for parametric animation. In contrast, parametric body models such as SMPL (Loper et al., 2023) and SMPL-X (Pavlakos et al., 2019) are widely used for realistic and animatable human bodies (Cao et al., 2024; Dong et al., 2024; Hong et al., 2022; Huang et al., 2023a;b; Pan et al., 2024; Wang et al., 2024a;b; Yang et al., 2024; Zhang et al., 2022; Dong et al., 2023; Xiu et al., 2022; 2023; Huang et al., 2024a; Liao et al., 2024). These models provide well-defined skeletons and consistent mesh topologies for easy animation. However, as shown in Figure 7 , they are designed for realistic human proportions and cannot represent the exaggerated features of anime-style characters.

In this paper, we present a novel pipeline that combines the controllability of SMPL-based models with the generative power of 3D diffusion models to synthesize high-quality anime-style 3D characters that are animatable. Unlike previous methods (Peng et al., 2024; Qian et al., 2025) that directly generate the entire character, our approach decomposes the character into modular components: body, hair, upper garments, lower garments, and accessories. Each part is generated as a separate high-resolution textured mesh for the final assembly.

To make the generated 3D anime characters animatable, we introduce Anime-SMPL, a parametric model designed for anime-style characters with high-quality meshes suitable for animation. Anime-SMPL retains the core SMPL structure while incorporating modifications for exaggerated eyes and anime-style proportions. Our Anime-SMPL also provides a consistent UV layout across characters, enabling direct texture generation in UV space. Conditioned on both a character image and a textual description, we adopt a multi-view diffusion model to predict UV textures for six semantic regions: body skin, facial skin, eyebrows, eyelashes, and the left and right eyes.

To generate meshes for hair, garments, and accessories, we propose a Multi-Shape Diffusion Transformer (DiT) architecture incorporating a Mixture-of-Experts (MoE) (Jacobs et al., 1991) module. This unified model dynamically routes inputs to specialized branches according to garment types to produce distinct meshes. We further guide the generation to be consistent with the body geometry by sampling points from the Anime-SMPL surface as an explicit geometry prior. As a result, our method produces well-aligned, separate meshes for hair, garments, and accessories.

In addition, we use a diffusion model to generate images of each garment component, conditioned on a reference image of the full body. Equipped with a multi-component self-attention mechanism, this model can selectively focus on relevant regions in the reference image. The resulting component images are then processed individually by an MVAdapter (Huang et al., 2024b) to produce the final textures of the garments.

The experiments demonstrate that our method achieves state-of-the-art performance in 3D anime character generation. Our method significantly outperforms previous approaches in both geometry and texture quality, while maintaining the highest fidelity to the input images.

Our main contributions are summarized as follows.

- We introduce Anime-SMPL, a unified human body template designed specifically for anime-style 3D characters. This template not only enables high-quality, animatable body mesh generation, but also facilitates direct texture synthesis in UV space.
- We propose a Multi-Shape DiT architecture incorporating a MoE mechanism that leverages character geometry as guidance. This design allows a single unified model to generate distinct meshes for each garment component while ensuring mesh compatibility with the underlying body shape, significantly reducing interpenetration artifacts.

- We propose a high-resolution, component-wise texture generation pipeline that employs a diffusion model to disentangle garment components from the input image. This enables independent texture synthesis for each component while mitigating color bleeding across different regions.

- Our method surpasses previous approaches in anime-style 3D character generation, achieving higher fidelity and overall quality. Moreover, our approach makes the generation of anime-style 3D characters practical for real-world applications, enabling downstream tasks such as garment retargeting, motion control, and facial expression control.

## 2 RELATED WORK

### 2.1 3D OBJECT GENERATION

In recent years, diffusion models have made remarkable breakthroughs in image generation. However, compared to massive 2D images, the scarcity of 3D data makes it challenging to directly train highly generalizable 3D generative models. To alleviate this issue, Poole et al. (2022) introduced the Score Distillation Sampling (SDS) loss, which leverages pre-trained 2D diffusion models to guide 3D generation. While SDS-based approaches (Chen et al., 2023; Lin et al., 2023; Poole et al., 2022; Qian et al., 2023; Raj et al., 2023; Tsalicoglou et al., 2024; Wang et al., 2023b;a) have significantly improved 3D generation performance, the resulting geometry often lacks fidelity, as SDS does not inherently enforce geometric consistency. Some methods (Lu et al., 2024; Long et al., 2024; Wu et al., 2024a) attempted to deal with this problem by reconstructing 3D geometry from multi-view normal maps, which can be inferred from a single image or synthesized by learned view generation, but they may suffer from limited volumetric consistency and difficulty in handling complex topologies.

With the increasing availability of 3D data, an emerging trend in recent methods is to move beyond SDS-based supervision. LRM-based approaches(Hong et al., 2023; Wang et al., 2024c; Li et al., 2023; Xu et al., 2024a;b; Tang et al., 2024; Zhang et al., 2024a;b) predicted triplane representations from limited inputs to reconstruct 3D models. Other methods (Wu et al., 2024b; Cui et al., 2024) applied diffusion models directly on 3D data to generate triplanes for 3D synthesis. In addition to triplane representations (Ren et al., 2024; Xiang et al., 2024), alternative formats such as sparse voxel grids (Xiang et al., 2024) and VecSet-based representations (Zhang et al., 2023; 2024c; Li et al., 2024; Chen et al., 2024b; Zhao et al., 2025; Li et al., 2025) are also gaining traction in 3D generation.

### 2.2 GARMENT GENERATION

Sewing patterns have been widely adopted in garment generation methods (He et al., 2024a; Liu et al., 2023) due to physical realism and alignment with realistic clothing construction. These representations provide clear semantic structure and high controllability, allowing for fine-grained adjustment of garment fit, style, and layout. However, generating accurate patterns requires expert knowledge, and may struggle with stylized or unconventional garments. Besides, inferring sewing patterns from images or text remains a challenge due to structural and alignment constraints. In addition to pattern-based generation, Qiu et al. (2023) reconstructed detailed 3D dynamic clothing from monocular video, Sarafianos et al. (2024) leveraged Long et al. (2024) to generate garments. Luo et al. (2025) proposed a body-aligned generation method of wearable assets based on native 3D diffusion. However, post-processing is still required after generation, since there are still small penetrations between the garment and body.

### 2.3 CHARACTER GENERATION

Previous works have achieved significant progress in realistic 3D human body generation. Xiu et al. (2022) leveraged implicit representations from normal maps to reconstruct detailed clothed human surfaces while Xiu et al. (2023) performd explicit normal integration for high-quality geometry recovery. Huang et al. (2023a) introduced Pixel-Semantics Difference-Sampling to optimize body and clothes, and Cao et al. (2024)introduced a dual-observation-space design consisting of a canonical space and a posed space related by a learnable deformation field for jointly optimization. Dong et al. (2024) proposed a layer-wise generation strategy to get better clothing structure and higher realness. Liao et al. (2024) further leveraged the synergy of a 2D diffusion model and a parametric body model

to generate digital characters. Several prior works have also explored 3D character generation in the anime style. Peng et al. (2024) used a large reconstruction model to generate anime-style 3D characters, but lack of details in many aspects. He et al. (2024b) and Dong et al. (2025) adopted a component-wise generation strategy to improve overall quality, and Qian et al. (2025) utilized sparse voxel to increase the resolution of generated meshes, enhancing quality of detailed regions like face and hands. However, due to limitations in fine-grained details, all these methods still fall short of producing truly usable 3D anime characters.

# 3 METHOD

Our pipeline, illustrated in Figure 2, first generates an image of the character in a canonical pose (e.g., A-pose) from either a textual description or an input image. Based on this canonical-posed image and the Anime-SMPL template, we estimate the Anime-SMPL parameters of the target character. Analogous to SMPL, we use a joint regressor to obtain joint locations. Subsequently, we sample points on the surface of the estimated Anime-SMPL mesh and encode them into body latent tokens using a VecSet VAE (Zhao et al., 2025). These body latent tokens, together with noised latent tokens, garment label tokens, and the canonical-pose image, are fed into our MoE-structured Multi-Shape DiT network to generate 3D garment meshes. Finally, we generate texture maps for both the body and individual garments.

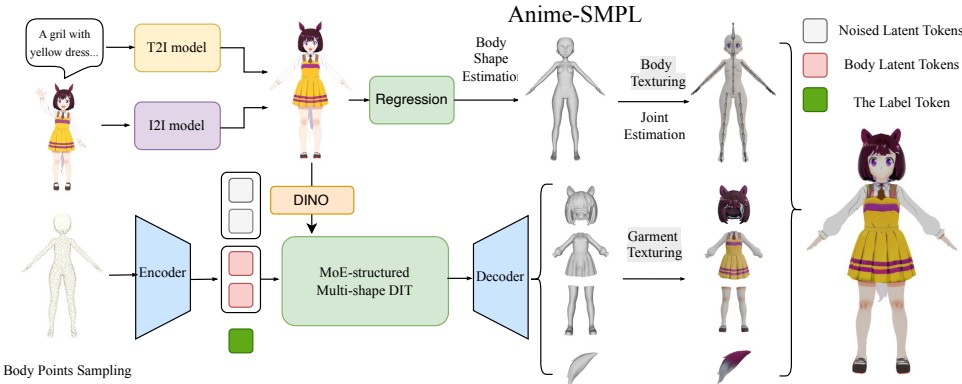

Figure 2: Pipeline for controllable 3D character generation based on Anime-SMPL and Body-Aligned Component-Wise Garment Modeling.

## 3.1 2D CANONICAL POSE CHARACTER GENERATION

As shown in Figure 2, our framework generates a front-view image of a character in a canonical pose from either a textual description or an image of the character in an arbitrary pose.

**Text-to-Image Synthesis.** We fine-tune PixArt-$\Sigma$ (Chen et al., 2024a) using paired data consisting of textual descriptions and corresponding front-view images of characters in a canonical pose.

**Image-to-Image Synthesis.** Similar to CharacterGen (Peng et al., 2024) and StdGen (He et al., 2024b), we employ a ReferenceUNet and a CLIP encoder (Radford et al., 2021) as image feature extractors, and incorporate pose information using a general a-pose skeleton image as an additional condition. To train our image-to-image model, we render diverse anime characters in full-body views under various camera viewpoints, poses, and facial expressions. To further improve the generalization performance of the model, we apply data augmentation strategies, including varying illumination conditions, modifying contour line styles, and upper-body-focused augmentations commonly used in pose estimation tasks (Xiao et al., 2018; Sun et al., 2019).

## 3.2 ANIME-SMPL

The SMPL model (Loper et al., 2015), has been instrumental in human shape estimation and 3D reconstruction, due to its ability to capture various body shapes and poses. However, the application of SMPL to anime characters is limited, owing to substantial geometric and stylistic differences, as illustrated in Figure 7. To overcome this limitation, we introduce Anime-SMPL, a parameterized body model tailored for anime-style characters, designed to better capture their unique proportions and stylistic characteristics.

**Anime-SMPL Parameterization.** In contrast to SMPL (Loper et al., 2015), which parameterizes both pose and shape, our framework focuses exclusively on shape modeling, as our aim is to reconstruct 3D anime characters from input images depicting a fixed canonical pose. As in SMPL, we apply Principal Component Analysis (PCA) to model shape variations in anime characters. We perform PCA on a dataset of 20,000 characters, each represented by 12,489 vertices, and retain the top 98 principal components to capture the dominant modes of shape variation. Similarly to SMPL, we estimate the joint regressor matrix $\mathbf{J}$ using non-negative least squares (NNLS), by solving the following optimization problem:

$$\underset{\mathbf{J} \geq 0}{\text{minimize}} \ \|\mathbf{J}\mathbf{V} - \mathbf{B_V}\|_F^2 \quad \text{s.t.} \quad \mathbf{J}\mathbf{1}_N = \mathbf{1}_K$$

where $\mathbf{V} \in \mathbb{R}^{N \times 3}$ is the vertex matrix, $\mathbf{B_V} \in \mathbb{R}^{K \times 3}$ is the target joint location matrix, $\mathbf{1}_N$ is an all-ones vector of length $N$, and $\mathbf{1}_K$ is an all-ones vector of length $K$. The problem is solved using the Splitting Conic Solver (SCS). The linear blend skinning (LBS) weights are directly adopted from pre-defined values in our dataset.

**Shape Parameters Estimation.** To estimate shape parameters for arbitrary anime characters, we train a shape prediction network on front-view images of characters exhibiting diverse clothing styles and body shapes. The network is optimized using the mean squared error (MSE) between the predicted parameters $\hat{\boldsymbol{\beta}}$ and the ground truth $\boldsymbol{\beta}$. In practice, we employ a ResNet-based architecture (He et al., 2016) as our shape prediction network.

## 3.3 MoE-STRUCTURED MULTI-SHAPE DiT FOR GARMENT GENERATION

Despite recent progress in 3D generation algorithms, synthesizing high-quality anime characters remains challenging. The difficulty arises primarily from the complex geometric structures inherent to anime characters, such as diverse hairstyles and intricate costumes. Previous approaches typically generate the entire character, making it difficult to precisely control fine-grained details in each part. To mitigate these challenges, we propose a component-wise generation strategy. Specifically, for each character, we decompose the outfit, excluding the body, into four components: *hairstyles*, *upper garments*, *lower garments*, and *accessories*. We employ the VecSet Diffusion Model (Zhang et al., 2023; 2024c; Li et al., 2024; Chen et al., 2024b; Zhao et al., 2025; Li et al., 2025) as the core generative model to synthesize these components.

**MoE-structured DiT Block.** The original VecSet Diffusion Model cannot directly generate garment components from a single image. To address this limitation, we design a MoE-structured Multi-Shape DiT architecture that enables independent generation of each garment component. Figure 3 illustrates the architecture of our MoE-structured Multi-Shape DiT.

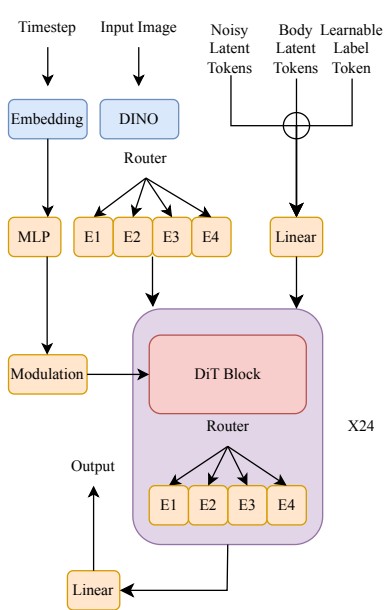

Figure 3: Overview of our MoE-structured Multi-Shape DiT.

In the VecSet Diffusion Model, the input image is first encoded by DINOv2 (Oquab et al., 2023) to produce conditioning tokens for the DiT backbone network. We adopt a Mixture-of-Experts (MoE)

design comprising four MLP expert branches, each dedicated to a specific garment component. In our MoE-structured Multi-Shape DiT, all parameters are shared except those of the four MLP experts, enabling precise, component-aware generation with minimal parameter overhead. We use a learnable label token to guide the network in generating the corresponding component.

**Body-Aligned Garment Generation.** Relying solely on image-based conditioning is insufficient for generating body-aligned garment components with the VecSet Diffusion Model, as it often leads to shape misalignments and noticeable interpenetration artifacts. In this section, we introduce an additional geometric condition to guide the shape and spatial alignment of the garment components.

Specifically, we perform point cloud sampling on the 3D body surface and encode it using the VecSet VAE encoder to obtain body latent tokens. The body latent tokens are then concatenated with the noised garment component tokens as input to the VecSet Diffusion Model. To reduce computational overhead, we encode the body latent tokens at lower resolution: the garment component tokens have a length of 3072, while the body latent tokens are set to 512. With this conditioning, which directly encodes spatial information of the body surface, the model is able to generate garment components accurately aligned with the underlying character geometry, which greatly reduces penetration issues and improves overall fidelity. The garment components are represented as Signed Distance Fields (SDFs) as output and are finally converted to 3D meshes via the marching cubes algorithm.

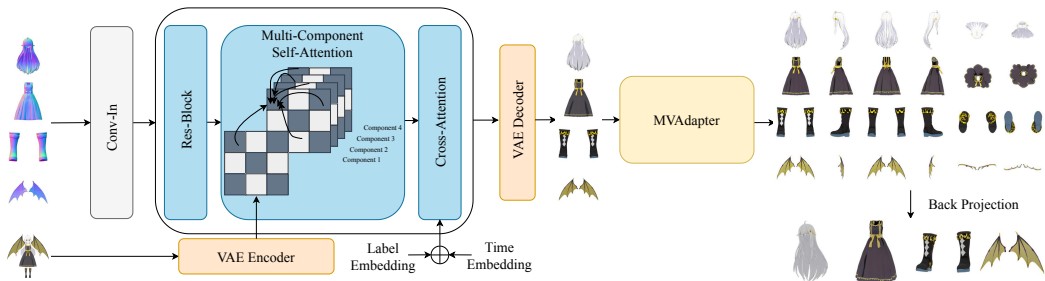

Figure 4: Pipeline of our Component-Wise High-Resolution Texture Generation.

## 3.4 TEXTURE GENERATION

Previous works, such as MVAdapter (Huang et al., 2024b), have demonstrated strong performance in multiview generation. Motivated by these advances, we adopt MVAdapter as a core component in our texture generation pipeline. Specifically, our method divides texture generation into two stages: one for the body and the other for the individual garment components.

**Body Texture Generation.** Leveraging the unified UV layout of Anime-SMPL across all characters, our method facilitates texture generation in UV space. We employ a multiview diffusion model (Huang et al., 2024b) as the texture synthesis network, which semantically decomposes the body texture into six regions: body skin, facial skin, left eye, right eye, eyebrows and eyelashes. The diffusion model takes region-specific text prompts as input and is conditioned on the character image to generate the UV texture for each region. A representative result is shown in Figure 5.

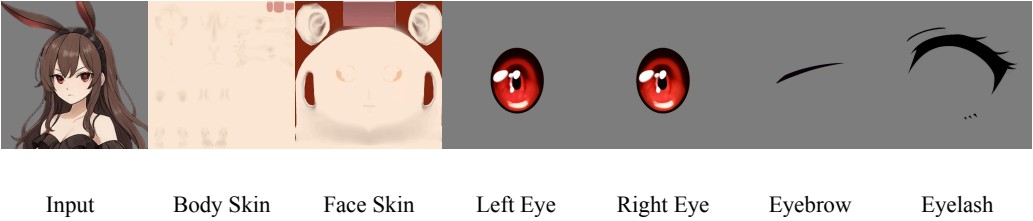

| Input | Body Skin | Face Skin | Left Eye | Right Eye | Eyebrow | Eyelash |

Figure 5: An example of the body texture we generated.

**Component-wise Garments Texture Generation.** Given the meshes of all garment components, we initially attempt to use normal maps as input, and employ MVAdapter (Huang et al., 2024b),

conditioned on the canonical-pose character image, to synthesize multi-view images for each garment component. However, this approach results in color bleeding, where the appearance of each component is adversely affected by neighboring regions.

To address this issue, we design a component-wise high-resolution texture generation pipeline, as illustrated in Figure 4. Following Dong et al. (2025), we first train a diffusion model to decompose the full-body image into enlarged and independent views of individual garment components. Specifically, we use the normal maps of all garment components as input and condition the model on the canonical-pose character image. A multi-component self-attention mechanism is employed to facilitate information exchange across components, and the resulting features are fused with label and timestep embeddings via cross-attention. This design enables the generation of high-quality images for these garment components. These segmented images are subsequently fed into the MVAdapter to synthesize multiview renderings, which are finally projected onto the 3D surface to obtain the complete texture map for each component.

Thanks to our component-wise garment generation strategy, self-occlusion is significantly alleviated when projecting textures from six canonical views (front, back, left, right, top, and bottom). Moreover, generating textures for each component allows for higher-resolution allocation, allowing for the generation of high-quality texture maps.

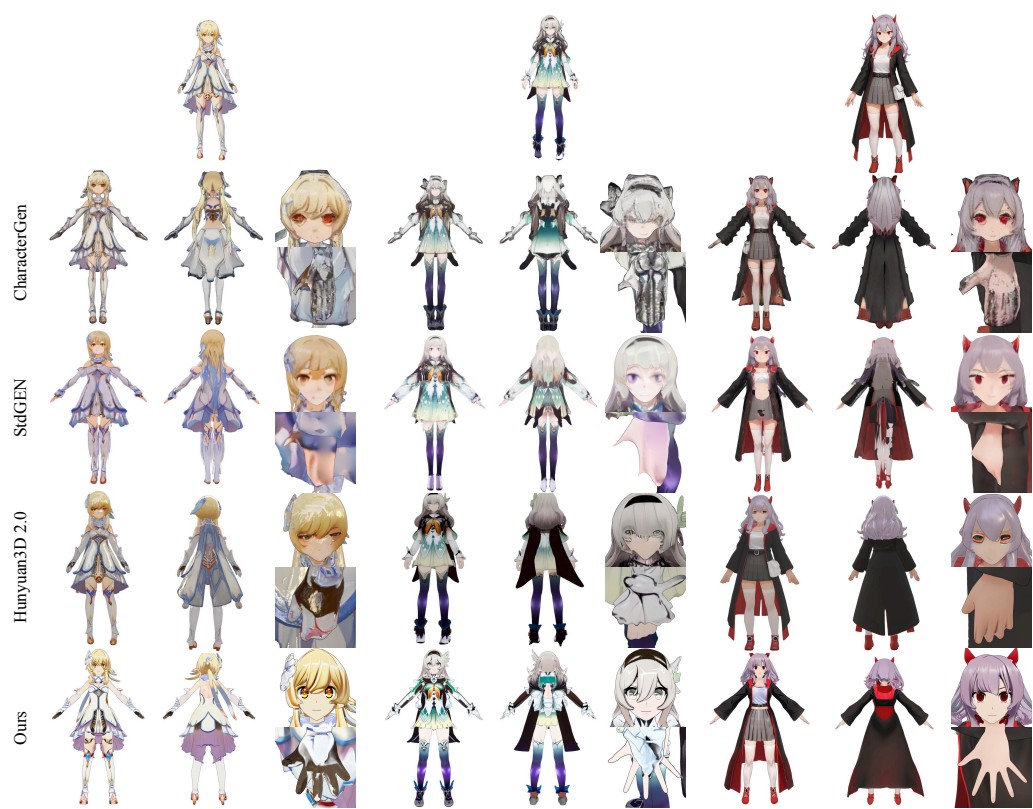

Figure 6: Qualitative comparisons between our method and previous state-of-the-art methods

# 4 EXPERIMENTS

## 4.1 IMPLEMENT DETAILS

**Training.** For the Anime SMPL shape prediction network, training is conducted on a single NVIDIA L20 GPU and takes approximately 4 hours. For the MoE-structured Multi-Shape DiT, we use 16 NVIDIA A100 GPUs with AdamW optimization and a learning rate of $1 \times 10^{-4}$, which requires about 10 days of training. For the remaining modules, 2D Canonical Pose Character Generation, Body

Table 1: User study results. The score ranges from 1 to 5, where 5 is the best score.

| Methods | Mesh Quality↑ | Texture Quality↑ | Fidelity↑ |
|---|---|---|---|
| CharacterGen (Peng et al., 2024) | 2.58 | 2.14 | 2.51 |
| StdGEN (He et al., 2024b) | 2.69 | 2.23 | 2.52 |
| Huanyuan3D 2.0 (Zhao et al., 2025) | 3.14 | 3.49 | 3.42 |
| Ours | **3.83** | **3.75** | **3.74** |

Texture Generation, and Component-wise Garment Texture Generation, each is trained independently on 8 NVIDIA A100 GPUs for approximately 2 days.

**Ineference.** The image generation stage takes 5 seconds, ANIME-SMPL parameter prediction takes 2 seconds, the MoE-Structured Multi-Shape DiT requires a total of 40 seconds, body texture generation takes 10 seconds, and garment texture generation takes 360 seconds.

**Dataset.** All models are trained on our private dataset, which consists of 20k anime characters aligned to a unified body template. Specifically, each character shares the same body mesh topology, including identical vertex count, vertex ordering, and face connectivity. For every character, garments are consistently divided into four components: hairstyle, upper garment, lower garment, and accessories. For characters missing any of these components, the corresponding part is left unassigned.

## 4.2 COMPARISONS WITH STATE-OF-THE-ART METHODS

To ensure a fair comparison with state-of-the-art anime-style character generation methods, we evaluate our model using both front-view images of characters in arbitrary poses collected from the Internet, as well as additional synthesized samples with diverse poses. The qualitative results are presented in Figures 6 and 12. These results demonstrate that our method produces highly detailed and realistic facial and hand regions of the characters.

Since our method and the baselines are trained on different datasets, we note that CharacterGen and StdGen are trained on Anime3D (Peng et al., 2024), whereas Hunyuan3D 2.0 is trained on large-scale datasets including ObjaverseXL (Deitke et al., 2023). In contrast, our method is trained on a proprietary dataset. Consequently, reconstruction-based metrics such as PSNR, SSIM, and LPIPS are not directly comparable. Instead, we conduct a user study that better reflects the perceptual quality of the generated results rather than simply computing losses against the ground truth. Specifically, we randomly sample 16 anime-style characters with diverse poses from the Internet as well as from synthesized images, and generate corresponding 3D models. Thirty participants are then asked to assess the visual quality of the generated textures and meshes, as well as their fidelity to the input images. To mitigate potential bias, participants are instructed to focus solely on the perceptual quality of the generated avatars, regardless of pose variations. As summarized in Table 1, the results show that our method consistently outperforms all competing approaches.

## 4.3 POSE CANONICALIZATION COMPARISON

The results of the pose canonicalization experiment are presented in Figure 13 (Appendix). Our method demonstrates superior generalization compared to prior approaches (Peng et al., 2024; He et al., 2024b; Zhao et al., 2025). We attribute this to several data augmentation strategies employed during training, including random cropping of the lower body, variations in lighting conditions, and adjustments to character contour line thickness.

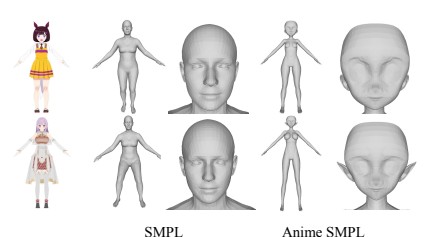

SMPL          Anime SMPL

Figure 7: Comparison between SMPL and Anime-SMPL on Anime Character Mesh Reconstruction

## 4.4 ABLATION STUDY

**Anime-SMPL vs. SMPL.** We compare the body mesh predicted by the SMPL model with that generated by our Anime-SMPL model. As shown in Figure 7, the mesh produced by our method aligns more closely with the input image in terms of ear shape, facial contour, and relative thigh and calf thickness. In contrast, the mesh predicted by the original SMPL model exhibits noticeable discrepancies in these areas. These results demonstrate that Anime-SMPL more effectively captures the distinctive geometric characteristics of anime-style characters.

**Body-Aligned Garment Generation.**

To verify the effectiveness of the proposed Body-Aligned Garment Generation strategy, we compare the garment-to-body alignment of models with and without body latent tokens, as illustrated in Figure 8. The results demonstrate that although our MoE-structured Multi-Shape DiT can roughly infer the spatial layout of each garment without explicit body geometry, the resulting garments often exhibit suboptimal alignment and increased susceptibility to interpenetration artifacts. In contrast, the incorporation of body latent tokens provides explicit geometric guidance, allowing the model to generate garments



Figure 8: Comparison of Garment Fitting with and without Body Latent Tokens.

that better conform to the body surface. This effect is especially pronounced for tight-fitting garments, such as swimsuits, where the generated clothing aligns precisely with minimal interpenetration.

**MoE-structured Multi-Shape DiT.** To demonstrate the effectiveness of the MoE design in our MoE-structured Multi-Shape DiT, we conducted a comparative experiment evaluating the generation performance of our MULTI-SHAPE DiT with and without MoE layers. As illustrated in Figure 9, the left shows the input image, the middle displays the upper garments generation results from DiT without MoE layers, and the right presents the up garments generation results from DiT with MoE layers. Our MoE-structured Multi-Shape DiT significantly outperforms the variant without MoE layers in terms of both generation quality and image-geometry alignment, thereby validating the effectiveness of our proposed MoE layer design.

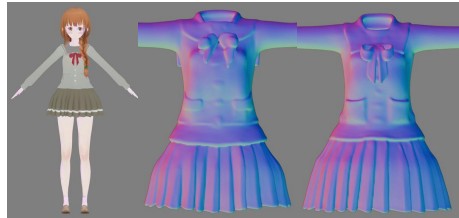

Figure 9: Comparison of the generation performance of our Multi-Shape DiT with and without MoE layers.

## 5 APPLICATION

Leveraging our Anime-SMPL model and the Body-Aligned Component-Wise Garment Modeling framework, the generated 3D anime characters can be directly applied to various downstream tasks, including garment retargeting, motion control, and facial expression control.

### 5.1 3D CHARACTER GARMENT RETARGETING

Leveraging the unified body template of our Anime-SMPL model, we facilitate straightforward and effective garment retargeting across characters. For each vertex on the source garment, we find its five nearest anchor points on the canonical body, compute a weighted relative displacement and apply it to the corresponding anchors on the target character to accurately transfer garment geometry. Sample retargeting results are presented in Figure 17.

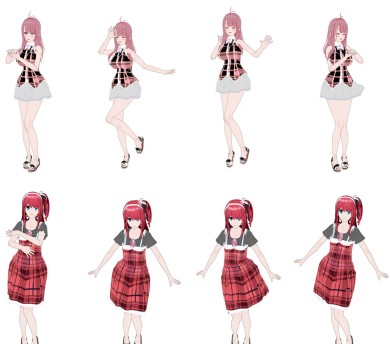

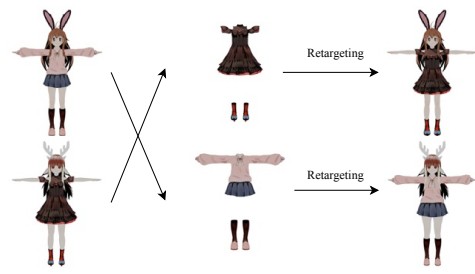

Figure 11: Garment Retargetting results.

Figure 10: Animation results.

## 5.2 MOTION CONTROL

Leveraging the precise skeletal structure and skinning weights of our Anime-SMPL model, high-quality motion control can be readily achieved. For garments without predefined skinning weights, we use a hybrid approach: body-hugging clothes inherit weights via nearest-neighbor sampling from the body, while garments with greater separation, such as skirts, are animated using physics-based simulation. Animation results are presented in Figure 1 and Figure 10. As illustrated in these figures, our generated 3D anime characters capture both natural full-body articulation and fine-grained movements, such as delicate finger gestures, thereby underscoring the precision and expressiveness enabled by our pipeline.

## 5.3 FACIAL EXPRESSION CONTROL

Leveraging the unified body template provided by our Anime-SMPL model, facial expressions can be controlled by manipulating facial vertices using blend shapes. Representative facial expression control results are shown in Figure 1. Furthermore, real-time facial expression control can be achieved via a face tracker, as demonstrated in Figure 18.

## 6 LIMITATIONS AND DISCUSSION

Although our method enables the generation of 3D anime characters from either text prompts or a single image, several limitations remain. First, the pose canonicalization step in the 2D Canonical-Pose Character Generation stage struggles with characters exhibiting complex poses or multiple accessories, often resulting in distorted outputs. Second, garment meshes extracted from SDFs via the marching cubes algorithm are inherently double-sided, which can impair physical simulation. Additionally, our texture generation using a multi-view diffusion model suffers from misalignment between projected images and the underlying geometry, as well as cross-view inconsistencies, which negatively affect texture quality.

For future work, mesh generation approaches (Chen et al., 2024c; Hao et al., 2024) that operate directly on vertices and faces may help mitigate the double-surface artifact. Moreover, generating textures directly in 3D space could alleviate cross-view inconsistency and enhance overall coherence.

## 7 CONCLUSION

In this work, we present a novel pipeline for generating high-quality, animatable anime-style 3D characters. We decompose each character into a set of componentsincluding body, hair, upper garments, lower garments, and accessorieswhich are generated independently. By integrating Anime-SMPL, a MoE-structured Multi-Shape DiT, and a high-resolution texture generation framework, our pipeline enables fine-grained modeling of both geometry and appearance. This design allows the production of high-quality, fully textured, animation-ready anime-style 3D characters.

ETHICS STATEMENT

This work complies with the ICLR Code of Ethics. Our research focuses on 3D anime character generation from text and images. The study does not involve human subjects, biometric data, or any sensitive personal information. We acknowledge the potential risks of misuse of generative technologies, such as producing inappropriate or misleading content. To mitigate these concerns, our research is restricted to academic purposes, and we encourage responsible use of our models aligned with ethical guidelines in the community. No conflicts of interest, sponsorship biases, or legal compliance issues are associated with this work.

ACKNOWLEDGEMENTS

This work is supported by the Hong Kong Research Grant Council General Research Fund (No. 17213925), National Natural Science Foundation of China (No. 62422606) and Gusu Innovation & Entrepreneurship Leading Talents Program (ZXL2024361).

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

## A  APPENDIX

## ACKNOWLEDGEMENTS

We sincerely thank the engineering team at DreamTech, especially Jingxi Xu, for their valuable assistance and support throughout this project.

We acknowledge the use of Large Language Models for assistance with language editing in this manuscript. All technical content and results are the sole responsibility of the authors.

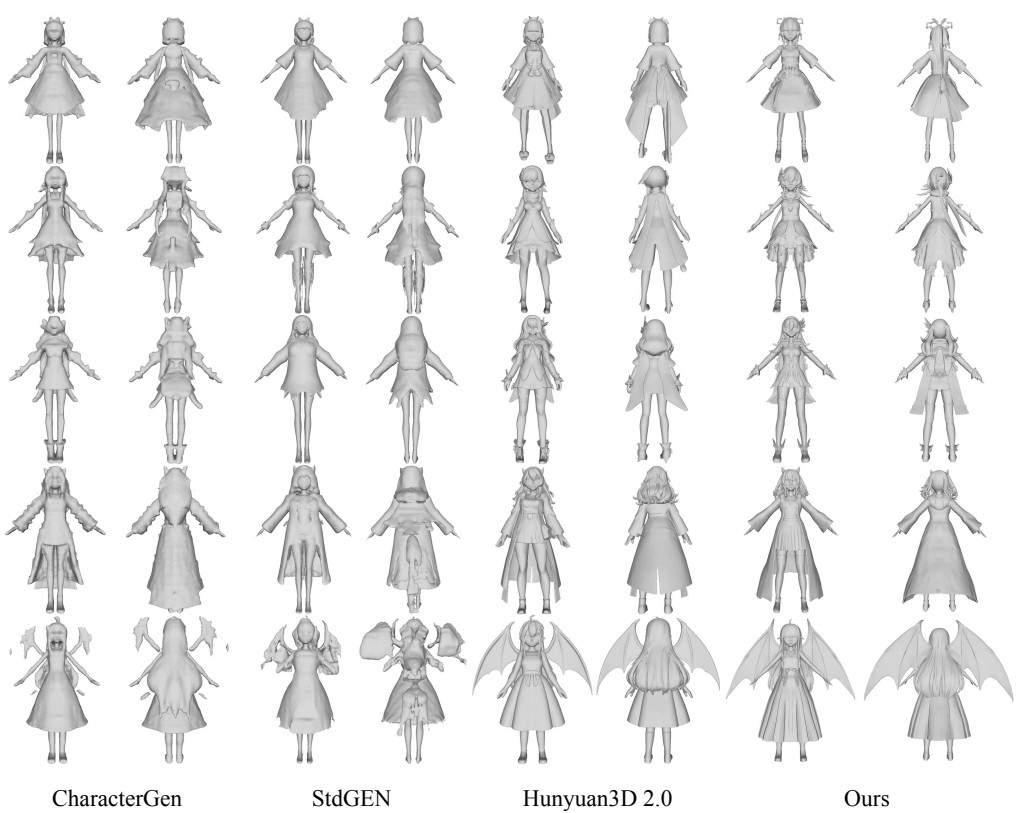

CharacterGen          StdGEN          Hunyuan3D 2.0          Ours

Figure 12: Mesh comparison between our method and other methods.

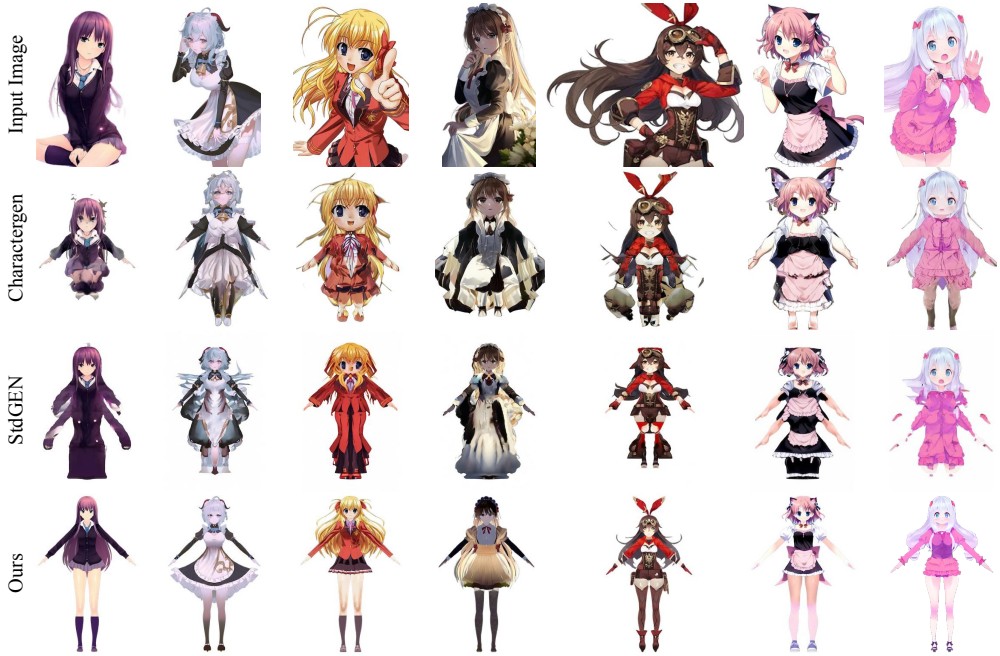

Figure 13: Pose Canonicalization Comparison between our method and other methods.

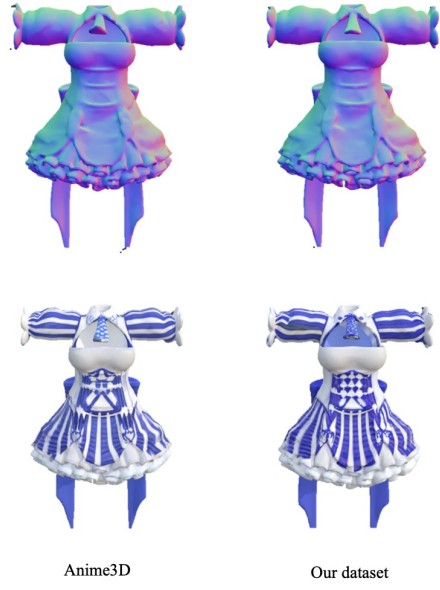

Anime3D          Our dataset

Figure 14: Since the Anime3D dataset does not have a unified body template, we can only train our component-wise garment model on it. As shown beyond, our method shows no significant difference in performance between the Anime3D dataset and our dataset.

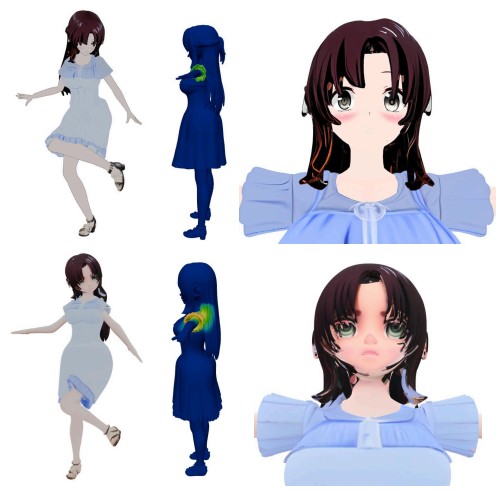

Figure 15: Comparison with Rodin. The first row shows our results, and the second row shows the results generated by Rodin followed by auto-rigging using Mixamo. The first column presents the animation results, and the second column shows the distribution of skinning weights for the shoulder bones.

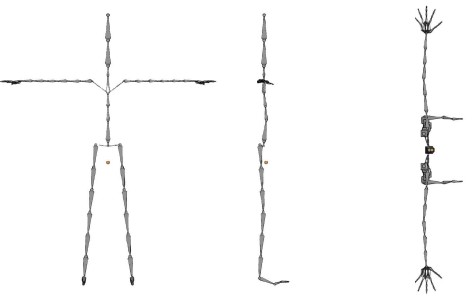

Figure 16: Visualization of our Anime-SMPL's joint structure.

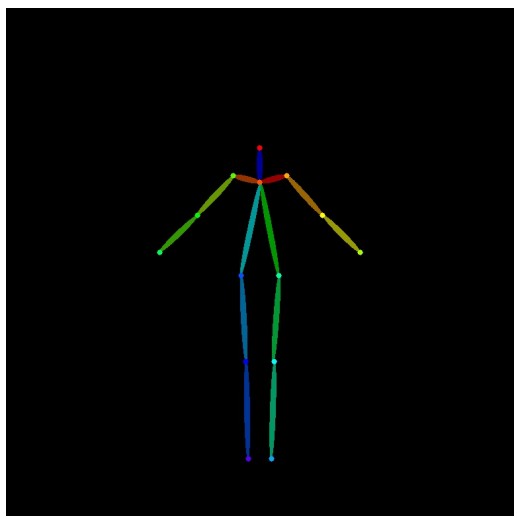

Figure 17: General a-pose skeleton image used in Image-to-Image Synthesis.

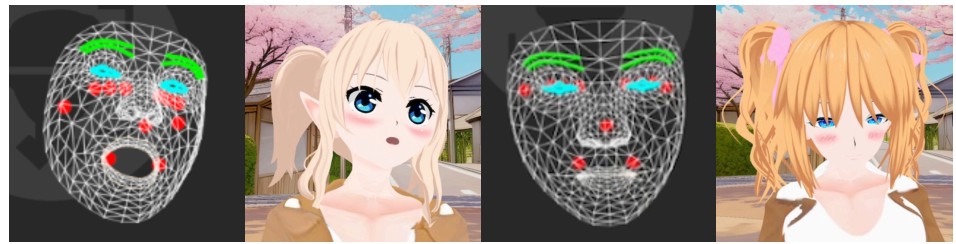

Figure 18: Real-Time facial expression control with face tracker.

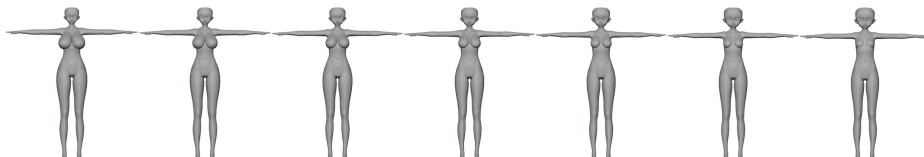

Figure 19: Visualization of the effect varying $\boldsymbol{\beta}$, we vary the first parameter of $\boldsymbol{\beta}$ from -2 to 2 from left to right.

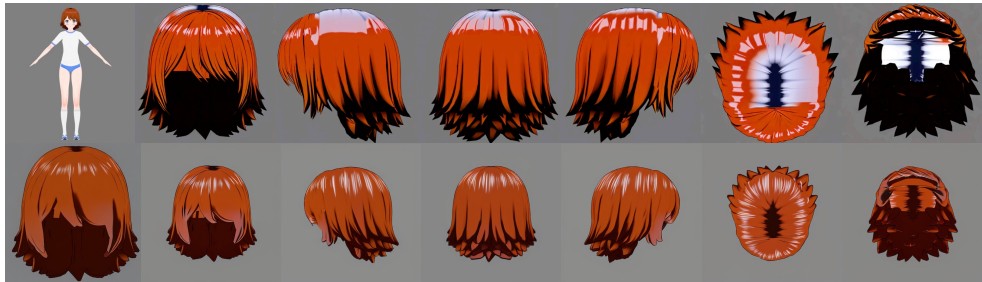

Figure 20: MVAdapter's Color Bleeding. The first row shows the input image on the far left and the result produced by MVAdapter on the right. In the second row, the far left displays the frontal view of the hair decomposed by our decompose model, while the right side shows the MV result obtained by using the decomposed image as input.

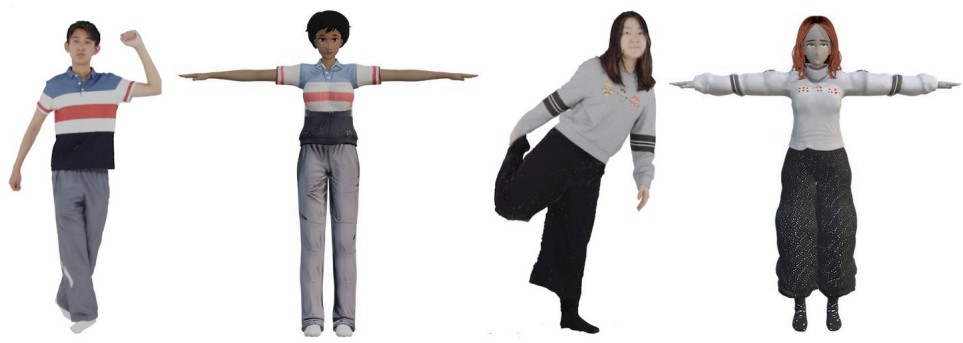

Figure 21: Our result on real humans.

