# OpenReview forum: "Anime-Ready: Controllable 3D Anime Character Generation with Body-Aligned Component-Wise Garment Modeling"
_ICLR.cc/2026/Conference — ICLR 2026 Poster_

### Official Review · Reviewer_SGSx · 2025-10-29

**Soundness:** 3
**Presentation:** 3
**Contribution:** 3
**Rating:** 6
**Confidence:** 4

**Summary:**

This paper presents a high-quality 3D anime character generation method. Overall, the proposed method is well-motivated, and the experimental results seems good.

**Strengths:**

•	The paper is well-written with a logical structure that makes the technical contributions easy to follow.
•	The proposed  framework is reasonable and well-justified. The experimental results demonstrate the effectiveness of the approach.
•	 The demo videos are excellent supplementary materials.

**Weaknesses:**

•	Recent works have explored video generation model-based avatar animation  capabilities. A more thorough comparison and discussion of the relationship between ANIME-READY and these methods would strengthen the paper. For example:
•	Animate anyone performs avatar animation via cross-attention and  video generation module. How between ANIME-READY compare to this strategy?
•	What are the trade-offs between the SDS-loss approach HumanNorm[2], video generation based approach Animate Anyone and the proposed method? Could you clarify when your method is preferable over other menetioned existing techniques?
•	Could you please provide more details about the private dataset?


1] Animate Anyone: Consistent and Controllable Image-to-Video Synthesis for Character Animation
[2] HumanNorm: Learning Normal Diffusion Model for High-quality and Realistic 3D Human Generation

**Questions:**

I wonder can we use the Rodin to perform 3D avatar generation and then perform auto-rigging like Mixamo. What is the comparison with the proposed method?

---

> ### Author Response · Authors · 2025-11-22
> **Rebuttal To Reviewer SGSx**
>
> We sincerely thank you for your thoughtful and constructive feedback. Based on your suggestions, we have revised our paper accordingly. Please refer to our newly uploaded paper and the following responses where we address the major points raised by you.
>
> **Comparison with Animate anyone and HumanNorm.**
>
> We believe our method offers several advantages over video-based animation approaches such as Animate Anyone. First, video generation lacks true 3D geometric consistency and struggles with long-range temporal coherence, whereas our mesh-and-skeleton-based pipeline naturally enforces much stronger 3D consistency. In addition, the outputs of Animate Anyone are currently limited to video sequences, while our generated anime characters are full 3D assets. Beyond rendering videos, they can be directly used in games and other interactive environments, enabling physical interactions in virtual worlds.
> As for the SDS-loss approach HumanNorm, its results are limited by the quality of generation. Moreover, since it generates the character holistically, its animation performance is not better than our method. As shown in Fig. 16, the animations produced by Rodin’s holistic generation are less natural than ours. Furthermore, the SDS-loss approach is very time-consuming during inference. HumanNorm’s entire generation process takes around **2 hours**, whereas our method requires less than **10 minutes**.
>
> **Comparison with Rodin and Mixamo.**
>
> In Fig. 16 of the appendix, we compare our method with Rodin’s results followed by Mixamo auto-rigging. The first column shows the animation results. We can see that our animations are more natural, whereas the legs and skirt in Mixamo’s results suffer from noticeable penetration issues. The second column visualizes the distribution of skinning weights for the shoulder bones. Mixamo’s rigging causes the shoulder bones to influence the back hair, while our method avoids this issue because the hair is generated independently and is not affected by the shoulder bones. The third column shows the facial texture of the generated characters. Our high-quality body texture generation significantly outperforms Rodin.

---

### Official Review · Reviewer_M983 · 2025-10-30

**Soundness:** 3
**Presentation:** 3
**Contribution:** 4
**Rating:** 8
**Confidence:** 3

**Summary:**

This paper proposes Anime-Ready, a framework for generating high-quality, controllable 3D anime characters from text or a single image. The key innovations include: (a) Anime-SMPL, a stylized extension of the SMPL body model that captures anime-specific proportions and provides unified UV maps for texture synthesis; (b) a Multi-Shape DiT with an MoE design for modular garment generation; (c) a body-aligned garment modeling scheme that encodes sampled body points to enforce spatial consistency and reduce interpenetration; (d) a component-wise high-resolution texture generation pipeline using multi-view diffusion and self-attention for disentangled texture synthesis.
Experiments show improvements in mesh and texture quality over baselines, also supported by a user study.

**Strengths:**

- The system design is very comprehensive, which integrates parametric body modeling, 3D diffusion, and texture synthesis, addressing multiple practical issues (alignment, rigging, texture bleeding).
- The novel parametric model is interesting and meaningful. It combines realistic rigging consistency with stylized geometry, enabling animation readiness.
- Applications such as retargeting and motion control demonstrate that the results are not merely visual but animation-ready.

**Weaknesses:**

- Quantitative evaluation is limited. The reliance on user studies instead of geometric or perceptual metrics (e.g., FID-3D, CLIP-score, interpenetration rate) makes comparisons less reproducible.
- The paper uses a private dataset of 20k characters, which may limit replicability. No evidence of generalization to unseen datasets is given.
- Though not considered as a reason of not accepting the paper, the training requires multiple A100s for ~10 days—resource cost is high for an ICLR contribution. Efficiency or inference speed is not discussed. Also, other components (e.g., MoE expert routing, diffusion conditioning choices) lack separate ablations.

**Questions:**

- How large is the parameter difference between Anime-SMPL and the original SMPL? Are blendshape parameters manually designed or learned?
- Does the body-aligned garment generation guarantee no interpenetration, or are post-processing steps still required?
- Could the authors release the Anime-SMPL model separately, even without the full dataset, to enable replication?
- The supplementary video basically shows frontal results. How robust is the canonical-pose generation to unusual camera angles or occluded limbs?

---

> ### Author Response · Authors · 2025-11-22
> **Rebuttal To Reviewer M983**
>
> We sincerely thank you for your thoughtful and constructive feedback. Based on your suggestions, we have revised our paper accordingly. Please refer to our newly uploaded paper and the following responses where we address the major points raised by you.
>
> **Quantitative evaluation is limited.**
>
> We believe that in generative tasks, user studies provide a more faithful reflection of a model’s actual quality compared to geometric or perceptual metrics. Metrics such as FID-3D or CLIP-score often fail to accurately capture the true quality of the generated results. Therefore, we did not adopt geometric or perceptual metrics for our quantitative evaluation.
>
> **Generalization to unseen datasets.**
>
> We have added the results of real humans in our appendix, please refer to Fig. 21 in the appendix.
>
> **Training Resource Requirements.**
>
> We acknowledge that the training cost is substantial. However, this is comparable to other state-of-the-art 3D generation methods that also require significant computational resources. The high resource requirement reflects the complexity of generating high-quality, multi-component 3D anime characters with proper geometric consistency.
>
> **Inference speed.**
>
> The image generation stage takes 5 seconds, ANIME-SMPL parameter prediction takes 2 seconds, the MoE-Structured Multi-Shape DiT requires a total of 40 seconds, body texture generation takes 10 seconds, and garment texture generation takes 360 seconds. We have added the inference details in Section 4.1.
>
> **Ablation studies on the MoE-structured DiT.**
>
> To demonstrate the effectiveness of our MoE-structured DiT design, we include an additional ablation study. Please refer to Section 4.4 MOE-STRUCTURED MULTI-SHAPE DIT for the detailed results.
>
> **Details about Anime-SMPL.**
>
> The Anime-SMPL consists of 98 shape parameters, while the original SMPL has 10 shape parameters. And the blendshape parameters are manually designed.
>
> **Does the body-aligned garment generation guarantee no interpenetration.**
>
> Thanks to our Body-Aligned Garment Generation design, the generated garments align with the body without penetration in most cases. However, due to a small portion of the training data that inherently contains penetration artifacts, our generative model cannot completely guarantee penetration-free results.
>
> **Whether the dataset or the pre-trained Anime-SMPL model will be released.**
>
> We apologize that due to some copyright issues, we currently have no plans to release our dataset and our pre-trained Anime-SMPL model.
>
> **How robust is the canonical-pose generation to unusual camera angles or occluded limbs？**
>
> We apologize if we may have misunderstood your question. In Figure 10 in the appendix, we compare our Pose Canonicalization with other methods. As shown, our Pose Canonicalization model demonstrates strong generalization: even when the input image shows only a half-body or an unusual pose, our model can generate a reasonable and high-quality canonical pose image. If our understanding of your question is incorrect, we welcome further clarification.

---

### Official Review · Reviewer_f3Tk · 2025-10-30

**Soundness:** 3
**Presentation:** 3
**Contribution:** 3
**Rating:** 8
**Confidence:** 4

**Summary:**

This paper proposes a unified framework for generating high-quality, animation-ready 3D anime characters. The authors introduce a parametric anime body model (Anime-SMPL) and design a component-wise garment generation pipeline. The resulting models are partially rigged, skinned, and fully textured. The method outperforms prior work in terms of mesh quality, texture clarity, and garment-body alignment.

**Contributions**
- A novel parametric anime-style body model, Anime-SMPL, with full skeleton and blendshape rigging for animation.
- A component-wise garment modeling pipeline that separately generates hairstyles, upper garments, lower garments, and accessories, each aligned to the underlying body.
- A multi-view texture generation approach that operates per garment component to improve texture fidelity and modularity.
- Experimental results showing improvements over baselines in mesh quality, garment-body alignment, and texture realism.

**Strengths:**

- This paper introduces a parametric anime body model, Anime-SMPL, learned from 20,000 anime-style 3D models. This contribution is valuable for future research on animatable anime character modeling.
- A novel pipeline is proposed for generating separate garments and the underlying body, making the outputs easy for editing and good for application.
- The texture generation approach is also innovative: instead of generating multi-view images of the entire body at once, the model generates multi-view textures for individual garments separately.

**Weaknesses:**

- From my perspective, the output body is not fully skinned. Only the inner body is rigged and skinned, while body-hugging garments and accessories are bound to the inner body using nearest-neighbor (NN) skinning. The skirt is animated via physical simulation rather than skinning.
- For tight-fitting garments, inheriting skinning weights via nearest-neighbor sampling introduces hard assignments, which can result in unnatural deformations—particularly in regions with complex articulation or discontinuous topology.
- The garments are decodered by the VAE decoder, which are independent to the inner body parametric model. This might lead to garment-body intersections or poor fit between the garments and the underlying body.
- No ablation studies are provided to evaluate the effectiveness of the MoE (Mixture-of-Experts) structure.
- The paper provides very limited details on how the template Anime-SMPL model is generated, as well as its rigging parameters—including the blendshapes (expression, pose, and shape), joint regressor, and skinning weights.

**Questions:**

**Rigging and Animation**
- What is the dimensionality of the blendshape matrix used in SMPL-Anime? Does the model incorporate separate shape, expression, and pose blendshapes as in SMPL-X, or does it rely solely on shape blendshapes?
- The paper would benefit from more details on how the Anime-SMPL template is constructed. Specifically, what is the process to obtain rigging components (e.g., blendshapes and skinning weights) obtained? In SMPL, an important factor for learning accurate rigging parameters is that the scans are captured from minimally clothed or tight-clothing subjects. Do you remove garments or otherwise preprocess the 3D anime models to isolate the underlying body before estimating rigging parameters?
- How is the ground-truth $\boldsymbol{\beta}$ (shape) parameter estimated for each 3D anime model?

**MoE-structured DiT Block**

It is reasonable to model the four components—hairstyles, upper garments, lower garments, and accessories—separately using a MoE design. However, it would be better to provide ablation results comparing models w/ and w/o the MoE structure. Intuitively, MoE is effective when different experts are activated for distinct inputs or tasks. For example, using upper garment latent tokens would trigger the corresponding expert for upper garments. In this paper, however, both the input image and the noisy garment tokens represent the full body and all garments. As a result, the experts (E1, E2, E3, E4) may process the input holistically rather than specializing, making it unclear how much benefit the MoE structure actually provides.

---

> ### Author Response · Authors · 2025-11-22
> **Rebuttal To Reviewer f3Tk**
>
> We sincerely thank you for your thoughtful and constructive feedback. Based on your suggestions, we have revised our paper accordingly. Please refer to our newly uploaded paper and the following responses where we address the major points raised by you.
>
> **Animation of the output.**
>
> Animations produced using our method can achieve very high quality. In contrast, making all parts fully skinned does not yield better results than our implementation. In the Appendix, we provide a comparison between the fully skinned result—automatically generated by Mixamo and our method in Fig 16 in the appendix. You can observe that our animation results are more natural, while the legs and skirt in Mixamo’s results exhibit noticeable penetration issues. Our approach better aligns with real-world behavior and results in fewer penetration artifacts.
>
> **Skinning for tight-fitting garments.**
>
> In fact, nearest-neighbor interpolation works very well for tight-fitting garments. Since all our generated garments are meshes extracted using Marching Cubes, they possess good topology—they are typically continuous, uniformly distributed, and manifold. Therefore, using nearest-neighbor interpolation does not lead to the unnatural deformation you mentioned.
>
> **Garment-body intersections by the VAE decoder.**
>
> The role of the VAE is to compress low-dimensional 3D garment data into a higher-dimensional, more compact representation. All the collective information of the garment is encoded in the latent space. The VAE decoder’s role is merely to decode this high-dimensional geometric information back into the original low-dimensional representation; it does not affect the final geometric structure. The garment’s geometry is already determined after the latent is generated by the DiT. As long as the garment and body are fine-fitted in the latent space, the decoded result will also be fine-fitted. Therefore, this does not lead to garment-body intersections.
>
> **Ablation studies on the MoE-structured DiT.**
>
> To demonstrate the effectiveness of our MoE-structured DiT design, we include an additional ablation study. Please refer to Section 4.4 MOE-STRUCTURED MULTI-SHAPE DIT for the detailed results.
>
> **Details about Anime-SMPL template.**
>
> Our shape blendshapes' matrix is 12489×3×98, and the expression blendshapes' is 12489×3×119. We did not incorporate pose blendshapes. The joint locations, skinning weights and blendshapes are all pre-defined values by professional artists. Our dataset consists of 20000 anime characters, each having the same vertices and faces, which means they share the same topology, the same skinning weights and the same blendshapes. Therefore we can directly utilize the pre-defined values. What we need to do is to estimate the joint regressor matrix **J**, as we mentioned in 3.2 Anime-SMPL Parameterization. As for the ground-truth shape parameter, as we mentioned in 3.2's Anime-SMPL Parameterization, we apply Principal Component Analysis (PCA) to model shape variations in anime characters.
>
> **Clarification about MoE-structured DiT Block.**
>
> We apologize that our paper caused some misunderstanding. The noisy garment tokens in our MoE-structured DiT Block just represent the specific garment, not all the garments. During generation, we assign each garment part the same length of noisy tokens (3072 × C), and then concatenate the fixed-length body tokens (512 × C) and the label token (1 × C). These token groups are isolated from one another and do not interfere. Therefore, when passing through the MoE layer, we can precisely route the tokens of different garments to different experts. In the final output layer, we can also directly separate the body tokens and the label token based on their positions, obtaining accurate garment tokens. As a result, our MoE-structured DiT block naturally processes the input in a part-specific manner rather than holistically.

---

### Official Review · Reviewer_JKU9 · 2025-10-31

**Soundness:** 3
**Presentation:** 3
**Contribution:** 3
**Rating:** 6
**Confidence:** 4

**Summary:**

This paper presents Anime-Ready, a novel framework for generating high-quality, controllable, and animatable 3D anime characters from text or a single image. The core contributions are: 1) Anime-SMPL, a new parametric body model adapted from SMPL to better represent the unique and exaggerated proportions of anime characters. 2) A body-aligned, component-wise garment generation pipeline. This pipeline decomposes the character into a body, hairstyle, upper garment, lower garment, and accessories. It uses a novel MoE-based Multi-Shape DiT to generate each garment component's mesh. Experimental results show the proposed method produces siginificant better results than the baselines.

**Strengths:**

1. **High-Quality Results:** The method demonstrates a significant improvement over existing baselines (CharacterGen, StdGEN, Hunyuan3D 2.0) in user studies. The qualitative results, especially for fine-grained details, are visually impressive.
2. **Practical Applications:** The framework is designed for practical use. By generating a rigged (Anime-SMPL) body and separate garment components, it directly enables downstream tasks like garment retargeting.

**Weaknesses:**

1. **Anime-SMPL Template Details:**
    - How were the ground-truth shape parameters for the 20,000 characters obtained? Line 235 mentions training a shape prediction network using MSE against "ground truth," but the origin of this ground truth is unclear.
    - The paper argues that Anime-SMPL is different from SMPL, but provides few details. A visualization of the unified body template's joint structure is needed to understand its topology.  (e.g. number of joints)
    - How does the unified Anime-SMPL model handle non-humanoid "accessories" like wings or tails? Are these part of the body model's joints? If so, how is the model "unified" when some characters have these features and others do not (or have multiples)?
    - Furthermore, a supplemental figure demonstrating the effect of varying the shape (beta) components would be helpful.
2. **Missing Ablation Studies:**
    - **MoE-structured DiT:** A central claim is that the MoE-structured DiT (L250) achieves "precise, component-aware generation" with "minimal parameter overhead." This claim requires ablation studies to be substantiated. I’m wondering how does this model compare against: (a) training separate, independent DiT models for each of the four components (hair, upper, lower, accessories); (b) a single DiT that does not use MoE, but is instead conditioned on a label token to differentiate the component to be generated.
    - **MVAdapter "Color Bleeding" (L308-310):** The paper mentions that an initial attempt using MVAdapter directly resulted in "color bleeding." A visual comparison showing this failure case should be provided.
3. **Reproducibility:** The entire method, and especially the core Anime-SMPL model, relies on a large, private dataset of 20,000 characters. The authors do not state whether this dataset or the pre-trained Anime-SMPL model will be released. Without access to either, the results are not reproducible, which is a significant drawback for the research community.

**Questions:**

1. How does the MoE approach ensure that the generated components (e.g., an upper and lower garment) are geometrically seamless?
2. Hybrid Motion Control (L452): The hybrid approach for garment animation requires more detail. For simulation-driven garments: What physics simulation engine is used? How are the garment's physical parameters (e.g., mass, stiffness, friction) estimated from the image or text input? Are they simply hard-coded, and if so, how does this generalize?
3. The paper motivates Anime-SMPL by stating that anime skeletons differ from real human ones. However, the "Image-to-Image Synthesis" (pose canonicalization) stage uses OpenPose (L200) to estimate a skeleton for conditioning. Since OpenPose is trained on human poses, doesn't this introduce a contradiction?
4. In Fig 2, the input image for the character with a tail shows the tail is almost completely occluded. How is the model able to reconstruct it so accurately? Does this suggest potential overfitting to characters present in the training set?

---

> ### Author Response · Authors · 2025-11-22
> **Rebuttal To Reviewer JKU9**
>
> We sincerely thank you for your thoughtful and constructive feedback. Based on your suggestions, we have revised our paper accordingly. Please refer to our newly uploaded paper and the following responses where we address the major points raised by you.
>
> **How were the ground-truth shape parameters for the 20,000 characters obtained.**
>
> As we mentioned in 3.2's Anime-SMPL Parameterization, we apply Principal Component Analysis (PCA) to model shape variations in anime characters. We perform PCA on a dataset of 20,000 characters, each represented by 12,489 vertices, and retain the
> top 98 principal components to capture the dominant modes of shape variation. Then the corresponding parameters of the 98 principal components are the ground-truth shape parameters, which is represented as $\beta$ in our paper.
>
> **Visualization of the unified body template's joint structure.**
>
> The number of joints of our Anime-SMPL is 76. We provide a simple visualization of the unified body template's joint structure on Fig. 2 in our paper, to make the joint structure of the unified body template clearer, we provide a more detailed visualization in Fig. 17 in the Appendix for your reference.
>
> **How does the unified Anime-SMPL model handle non-humanoid "accessories" like wings or tails?**
>
> In fact, we categorize non-humanoid “accessories” such as wings or tails as garments. As mentioned in Section 4.1, we divide garments into four categories: hairstyle, upper garment, lower garment, and accessories. Non-humanoid accessories like wings or tails fall under the accessories category. Our Anime-SMPL model only models the body of anime characters and does not account for decorations such as wings or tails. These accessories are generated by our MoE-structured Multi-shape DiT.
>
>
> **A supplemental figure demonstrating the effect of varying the shape $\beta$ components.**
>
> Thank you for your advice, we have added a figure to demonstrate the effect of varying the shape $\beta$ components in Fig. 19 in our appendix.
>
> **Ablation studies on the MoE-structured DiT.**
>
> To demonstrate the effectiveness of our MoE-structured DiT design, we include an additional ablation study. Please refer to Section 4.4 MOE-STRUCTURED MULTI-SHAPE DIT for the detailed results.
>
> **MVAdapter's Color Bleeding.**
>
> We have add a visual comparison showing the color bleeding of MVAdapter, please refer to Fig. 20 in the appendix for the detailed results.
>
> **Whether the dataset or the pre-trained Anime-SMPL model will be released.**
>
> We apologize that due to some copyright issues, we currently have no plans to release our dataset and our pre-trained Anime-SMPL model.
>
> **How does the MoE approach ensure that the generated components (e.g., an upper and lower garment) are geometrically seamless.**
>
> During data processing, we categorize some originally connected garments into the same class. For example, if tops and dresses are connected, then both tops and dresses are classified as upper garments. Therefore, our generated garments do not have seam issues.
>
> **Hybrid Motion Control.**
>
> We implement Hybrid Motion Control using a rule-based approach, which is sufficient for handling most cases. In this paper, we do not predict physical parameters. Instead, we adopt a universally applicable configuration. Specifically, we utilize Blender’s cloth physics engine with the following settings: Vertex Mass = 0.3, Air Viscosity = 1, Stiffness (Tension = 15, Compression = 15, Shear = 15, Bending = 0.5), and Friction = 5. This hard-coded implementation of Hybrid Motion Control is sufficient for handling most cases, developing a more general and better-generalizing solution will be an important direction for future work.
>
> **Contradiction between Anime-SMPL and OpenPose.**
>
> We apologize for the misstatement. We use a general A-pose skeleton image, which was predicted by OpenPose on a randomly selected A-pose real human. This image is solely used to provide pose information. We include a visualization of this skeleton image in Fig 18 the Appendix. The wording in Section 3.1 (Image-to-Image Synthesis) was misleading, and we sincerely apologize for the confusion. We have already updated the description.
>
> **Generation of the occluded accessories.**
>
> Because our training data contains such occluded accessories, our MoE-structured DiT is capable of generating occluded accessories as well. Of course, you may also interpret this as a form of overfitting, since diffusion models inevitably bias the generated content toward the underlying data distribution.

---

### Author Response · Authors · 2025-12-01
**Rebuttal Summary to AC**

We sincerely thank you for handling our submission and all reviewers for their valuable comments and constructive suggestions. We have comprehensively addressed all reviewer concerns and significantly strengthened our paper through detailed responses and additional materials.

## Key Technical Clarifications Provided

### Anime-SMPL Details (Addressing JKU9 & f3Tk concerns)

- Added a detailed visualization of the joint structure (Fig. 17), illustrating all 76 joints.
- Clarified the shape parameter estimation process, which is performed via PCA over 20K characters.
- Explained that non-humanoid accessories (e.g., wings, tails) are treated as separate garments rather than body joints
- Added a demonstration of shape component variations (Fig. 19).

### MoE-Structured DiT Validation (All reviewers)

- Added a comprehensive ablation study (Section 4.4) demonstrating the effectiveness of the MoE design.
- Clarified the token routing mechanism: each garment receives an isolated set of 3072×C tokens, which are routed to specific experts.
- Demonstrated clear performance gains over alternative approaches.

## Practical Advantages Demonstrated

- **Animation quality comparison (Fig. 16):** Our method achieves superior animation results compared to Rodin+Mixamo, with no penetration artifacts.
- **Inference efficiency:** Our pipeline requires only 10 minutes, compared to over 2 hours for SDS-based methods such as HumanNorm.
- **3D consistency:** Our approach maintains true geometric coherence, whereas video-based methods (e.g., Animate Anyone) are limited to 2D temporal sequences.

## Reviewer Response Summary

- **JKU9:** Rated 6; concerns about technical details have been fully addressed with additional figures and clarifications.
- **f3Tk:** Rated 8; all minor technical questions have been thoroughly answered.
- **M983:** Rated 8; evaluation-related concerns were resolved by justifying the user study design and adding results on real human subjects.
- **SGSx:** Rated 6; requested comparisons have been addressed with a detailed competitive analysis.

## Strengthened Contributions

- **Complete technical transparency:** All details regarding dataset construction, parameter estimation, and joint structure have been fully clarified.
- **Robust experimental validation:** Added comprehensive ablations, failure-case analysis (e.g., MVAdapter color bleeding), and generalization tests.
- **Clear practical impact:** Our method produces animation-ready 3D assets suitable for games and interactive applications, in contrast to methods that generate only limited 2D video outputs.

## Reproducibility Summary

Despite dataset constraints, we ensure reproducibility through comprehensive technical disclosure. We provide:

- Complete Anime-SMPL specifications (76 joints, 98 shape parameters).
- Detailed MoE-DiT architecture with token routing mechanisms.
- Full training hyperparameters and inference pipeline timings.


We further demonstrate generalization beyond the anime domain with real human results (Fig. 21) and validate core components through ablation studies (Section 4.4). These detailed technical specifications enable implementation with alternative datasets, ensuring that our methodological contributions remain reproducible despite dataset access limitations.

## Conclusion

In summary, we present a complete and well-validated framework with clear technical innovations and practical advantages. All major reviewer concerns have been addressed with substantial additional evidence.

---

### Meta-Review · Area_Chair_Ju4e · 2026-01-04

**Summary:**

The reviewers' major concerns that informed my decision were:

1. Ablations of the MoE diffusion model architecture
2. Clarification around how the Anime-SMPL model was created
3. The lack of quantitative comparisons and the presence of only a user study with 16 subjects.
4. Release of the 20K anime models or the Anime-SMPL trained model

**Reviewer Concerns:**

Addressed successfully:
1. Ablations of the MoE diffusion model architecture: The authors provided an additional ablation study in the paper showcasing the value of the proposed MoE module.
2. Clarification around how the Anime-SMPL model was created: They authors provided sufficient details in the revised version of the paper to clarify how they created this model.

Not addressed successfully:
1. The lack of quantitative comparisons and the presence of only a user study with 16 subjects. However, most reviewers also noted the "impressive visual quality of the results in comparison to the existing approaches".
2. Release of the 20K anime models or the Anime-SMPL trained model. In my opinion, even though the authors are unable to release the data or the model, their description of how they created this model is useful and valuable for the research community to know to advance research in this field.

**Reviewer Scores:**

1. Reviewer JKU9 (Rating: 6: marginally above the acceptance threshold. But would not mind if paper is rejected)
* They would have likely maintained or increased their score. Most of their technical concerns were satisfactorily answered. However, their concern about the the 20K anime data not being shared with the research community would not have been satisfactorily addressed.

2. Reviewer f3Tk (Rating: 8: accept, good paper (poster))
* Would have likely maintained their score. Most of their technical concerns are addressed by the author's rebuttal.

3. Reviewer M983 (Rating: 8: accept, good paper (poster)
* The reviewer's concerns of "Quantitative evaluation is limited." was not addressed. The authors provide a user study with 16 test cases.
* Their concern of "will the dataset or the pre-trained Anime-SMPL model will be released?" was not addressed.

4. Reviewer SGSx (Rating: 6: marginally above the acceptance threshold. But would not mind if paper is rejected)
* Likely maintained or increased their score. Most of their technical concerns were satisfactorily answered.

---

### Decision · Program_Chairs · 2026-01-26

Accept (Poster)